# Evaluation of the Perceptual Interactions among Aldehydes in a Cheddar Cheese Matrix According to Odor Threshold and Aroma Intensity

**DOI:** 10.3390/molecules25184308

**Published:** 2020-09-19

**Authors:** Chen Chen, Wenya Zhou, Haiyan Yu, Jiajie Yuan, Huaixiang Tian

**Affiliations:** Department of Food Science and Technology, Shanghai Institute of Technology, Shanghai 201418, China; chenchen@sit.edu.cn (C.C.); zwyxu0908@163.com (W.Z.); hyyu@sit.edu.cn (H.Y.); yjj18964018381@163.com (J.Y.)

**Keywords:** cheddar cheese, nutty flavor, cheese matrix, threshold, synergism

## Abstract

To evaluate the contributions of 3-methylbutanal, 2-methylbutanal, 2-methylpropanal, and benzaldehyde in cheddar cheese models, the threshold values, optimal concentration ranges, and perceptual actions of these compounds were determined at various concentrations. The thresholds for 3-methylbutanal, 2-methylbutanal, 2-methylpropanal, and benzaldehyde in the cheese matrix were 150.31, 175.39, 150.66, and 500.21 μg/kg, respectively, which were significantly higher than the corresponding values in water. The optimal concentration ranges of these aldehydes were determined as 150–300, 175–325, 150–350, and 500–1500 μg/kg, respectively. Based on the results of the threshold method and Feller’s model, five binary mixtures were found to have synergistic effects, and only the pair of 2-methylpropanal and benzaldehyde was determined to have a masking effect. In addition, the synergistic olfactory effects between the four ternary mixtures and the quaternary mixture of these aldehydes were also assesSsed using Feller’s model. In a σ-τ plot analysis, synergism was usually observed when these odor pairs were at their threshold levels. In summary, the results suggested that perceptual interactions among these aldehydes exist in a cheese model variably with different concentrations and threshold ratios. This study will be helpful to a further understanding of the nutty aroma and improving the aroma quality of cheddar cheese.

## 1. Introduction

Cheddar is a hard cheese produced by the gelatinization of milk with rennet, followed by acidification and concentration [1]. It is the most abundant cheese in the world in terms of production and overall consumption [2]. Aroma is an important sensory property of cheese and is one of the first stimuli to be perceived before consumption [3,4]. Among the flavor features, a nutty flavor is typically preferred by most consumers of cheddar cheese [5].

To date, more than 600 volatile compounds have been identified in Cheddar cheese [6]. Many previous studies have reported that some of these substances, such as methyl pyrazine, 2,5-dimethyl pyrazine, 2,6-dimethyl pyrazine, trimethyl-pyrazine [7], 2-methylbutanal, 3-methylbutanal, and 2-methylpropanal contribute to the nutty flavor of cheese. Of these compounds, 3-methylbutanal, 2-methylbutanal, and 2-methylpropanal are regarded as the primary sources of nutty flavors in cheddar cheese [8,9,10,11]. In addition, we previously determined that benzaldehyde contributes to the nutty flavor of cheese, as perceived by Chinese consumers [12].

The olfactory threshold is known to be an important indicator of the flavor contribution of a compound. To evaluate the influences of individual compounds on aroma, the concentration of each compound is divided by the respective odor threshold to calculate the odor activity value (OAV) [13]. The thresholds of 2-methylbutanal, 3-methylbutanal, 2-methylpropanal, and benzaldehyde in air, water, and oil systems have previously been reported [14]. In addition, thresholds of 2-methylbutanal, 3-methylbutanal, 2-methylpropanal, and benzaldehyde are also determined in model wine solution [15]. However, threshold values are dependent on the matrix and therefore vary widely in different matrices [16,17]. To the best of our knowledge, threshold values for these compounds in cheese have not been reported. Furthermore, suitable concentrations of aroma compounds may impart a desirable flavor to a food, whereas excess amounts of these compounds may lead to off-flavors and reduce consumer acceptance. For example, high concentrations of 3-methylbutanal have been reported to produce off-flavors in raw milk [18,19]. Accordingly, the desirable nutty flavor in cheddar cheese suggests that major flavor compounds are present at appropriate levels such that they do not impart off-flavors.

It has been reported that the chemical complexity of an odorant mixture influences the nature of perceptual interactions [20,21,22]. Aroma synergy generally means that the aroma perception intensity of the mixture is greater than the sum of the aroma perception intensities of each component in the mixture. For example, suppose that compound A has an aroma intensity of x at a concentration of n, compound B has an aroma intensity of y at a concentration of n, and the mixture of compounds A and B has an aroma intensity of z at a concentration of 2n. When z > x + y, synergy occurs [23,24,25]. At present, methods to evaluate aroma synergy mainly include the threshold method, Feller’s additive model, OAV determination, and the σ-τ diagram method [20]. These methods mainly determine the synergy between components through changes in the threshold value, OAV, and aroma intensity before and after the combination of aroma components. Lytra et al. [24] used Feller’s additive model to study the interactions between esters in red wine and found that ethyl-3-hydroxybutanoate and 2-methylpropyl acetate led to a significant decrease in the threshold of the fruit pool in red wine, thus demonstrating the synergistic effects of these compounds in increasing the overall aroma intensity. Zhu et al. [26] used the threshold method to show that 3-methylbutanal and 2-methylbutanal have a synergistic effect in oolong tea because of their similar structures and aromas. We have previously used Feller’s additive model and σ-τ diagrams to study the synergistic effect of key aroma compounds in traditional Chinese cheese [27]. However, reports on the synergistic effects of flavor compounds in cheddar cheese are scarce.

In this study, we first evaluated the odor thresholds and then determined the suitable concentration ranges of 3-methylbutanal, 2-methylbutanal, 2-methylpropanal, and benzaldehyde in a cheese matrix. Finally, we evaluated the perceptual interactions of these four compounds using the threshold method, Feller’s additive model, and σ-τ diagrams. Different degrees of synergism were seen among these aldehydes, depending on their thresholds and concentrations. Accordingly, the phenomenon of synergism provides additional theoretical support for techniques to improve the flavor quality of cheddar cheese.

## 2. Results and Discussion

### 2.1. Thresholds of Aldehydes in the Cheese Matrix

Table 1 lists the determined and referenced threshold values of these compounds, together with descriptions of the aromas. The detected thresholds of 3-methylbutanal, 2-methylbutanal, and 2-methylpropanal in the cheese matrix were 150.31, 175.39, and 150.66 μg/kg, respectively, higher than the corresponding values detected in water (*p* < 0.05). An exception to this finding was benzaldehyde for which the threshold values were of the same magnitude in the cheese matrix (500.21 μg/kg) and water (350 μg/kg). These results are consistent with those of studies on beer aromas performed by Meilgaard et al. [28] in which the odor threshold values of most of the tested volatile compounds were higher in beer than in water. For example, the odor threshold values for butanol, 3-methylbutanol, and dimethyl sulfide were higher in beer samples by factors of 400, 280, and 152, respectively. The thresholds of 2-methylbutanal and 3-methylbutanal in the wine solution model are 16 and 4.6 μg/L, respectively [15]. The thresholds for two compounds differed significantly in the wine solution model and the cheese model. These differences in threshold values may be attributed primarily to the food matrix [17]. The complex macronutrients in cheese (i.e., fat, protein, and carbohydrate) may mask the expression of volatile compounds [29].

### 2.2. The Optimal Concentration Ranges of Aldehydes in a Cheese Matrix

The optimal concentration ranges of the four aldehydes in the cheese matrix were determined. As shown in Figure 1, the optimal concentration ranges for 3-methylbutanal, 2-methylbutanal, 2-methylpropanal, and benzaldehyde were 150–300, 175–325, 150–350, and 500–1500 μg/kg, respectively. The variation of aroma compound concentrations within a mixture to produce different aroma types can also change the flavor character [30]. For example, the nutty aroma intensity was enhanced with increasing 3-methylbutanal concentrations ranging from 150 to 300 μg/kg. When the concentration exceeded 300 μg/kg, the sensory character of 3-methylbutanal changed from being predominantly a nutty aroma to being dominated by an unpleasant grass odor. This is in accordance with previous studies that identified 3-methylbutanal as fruity and pleasant at low concentrations, but as causing unclean-harsh and dulling flavor sensations when present at concentrations greater than 200 μg/kg in cheddar cheese [31]. The optimal range of 3-methylbutanal determined in this study was higher than previously reported values, probably due to the different matrices used to determine the optimal concentrations [29]. The result of 2-methylpropanal was similar to that of 3-methylbutanal and the overall flavor scores hardly exceeded value of six. As to the 2-methylbutanal, when its concentration increased from 175 to 325 μg/kg, the scores of nutty flavor and overall flavor also gradually increased. Especially when the concentration of 2-methylbutanal reached 325 μg/kg, the scores of nutty flavor and overall flavor were highest (7.2 points) among all the concentrations tested. But when its concentration reached 375 μg/kg, the overall flavor score sharply decreased to 4.7 points. This may be ascribed that although 2-methylbutanal can improved the nutty flavor, but it also brought a green malty flavor to the cheese when its concentration exceeded a limit. The phenomenon observed for benzaldehyde was somewhat different. When its concentration in the cheese matrix exceeded 1500 μg/kg, the intensity of the nutty aroma was strong, but the overall aroma was sharp for panelists.

### 2.3. Evaluation of the Perceptual Interactions among Aldehydes in the Cheese Matrix Using the Odor Threshold

The threshold values of compounds indirectly reflect their influences on aroma. Thus, the overall aroma of a mix of compounds could be considered as the sum (at least approximately) of the aroma of each single compound. However, this assumption neglects the interactions between compounds, which differ from those seen in model cheese conditions [25]. Therefore, using the threshold values as a reference, we compared the variations in threshold values of the compounds before and after being mixed to evaluate the relationships among compounds [26,32].

Based on data from the literature, compounds were added to the cheese model in proportion to the individual thresholds detected in the present study [33]. The ratio (%) of the determined threshold value of a compound in the mixture to the original threshold value of the individual compound was used to represent the degree of practical interaction. In this experiment, six binary compound combinations were tested.

As shown in Table 2, when compounds with similar structures and aromas were mixed, a synergistic effect was observed on the aroma of the mixture. For example, 2-methylbutanal and 3-methylbutanal are isomers. 2-Methylbutanal produced malty, cacao, and apple-like aromas, while 3-methylbutanal produced malty, coffee, and cacao aromas. Therefore, the mixture of these compounds emitted a pleasant nutty and malty odor with a threshold of 29.29% of the single compound thresholds, thus showing a strong synergistic effect. This is consistent with previous studies that demonstrated the synergistic effects of these two compounds in oolong tea [26] and beer [33]. Furthermore, the ratios of 2-methylbutanal to 2-methylpropanal and 2-methylpropanal to 3-methylbutanal were 49.08% and 40.90%, respectively, which indicated synergistic actions. However, a masking effect was observed among compounds with different structures. For example, the ratio of 2-methylpropanal to benzaldehyde was 148.20%. The above results indicate that the threshold value does change when the two substances are mixed due to the fact of their different perceptual interactions.

### 2.4. Analysis of Interactions Using Feller’s Additive Model

To better understand the synergistic effects of the nutty aroma compounds of cheddar cheese, the detection proportion of each compound was calculated in each binary mixture using Feller’s additive model [34]. The four nutty aroma compounds produced six binary mixtures, each of which was mixed at the measured threshold ratio. As shown in the S-curves for (a) 2-methylbutanal and 3-methylbutanal, (b) 2-methylbutanal and 2-methylpropanal, (c) 3-methylbutanal and 2-methylpropanal, (d) 3-methylbutanal and benzaldehyde, (e) 2-methylbutanal and benzaldehyde, and (f) 2-methylpropanal and benzaldehyde mixtures, presented in Figure 2; the five measured binary mixture thresholds were below the thresholds calculated using Feller’s additive model when the detection probability was 50%. The ratios of the calculated to measured thresholds were 3.24, 2.21, 2.45, 2.69, 1.02, and 0.89, respectively, for the aforementioned pairs of compounds. Especially for the pair of 2-methylbutanal and 3-methylbutanal, compounds with similar structures and aromas yielded the strongest synergistic effects. However, the threshold value of the mixture of 2-methylpropanal and benzaldehyde was higher than the theoretical threshold value, indicating a masking effect. These results are consistent with those obtained using the threshold method. This may be due to the cyclic structure of benzaldehyde in which an aldehyde replaces a hydrogen [35], while 2-methylpropanal is a branched-chain aldehyde. In addition, benzaldehyde has an almond aroma, but 2-methylpropanal has a varnish aroma. The differences in the structures and aromas of these compounds may result in a masking effect in the mixture. The results are consistent with previous report results that compounds with similar structures (homologs) have obvious synergistic or additive effects, and little interaction was found among compounds with different structure and aroma [33].

To further explore the perceptual interactions among these aldehydes, the overall perceived odor intensities of the ternary mixtures and quaternary mixture were analyzed. As shown in the S-curves presented in Figure 3, the thresholds of all the measured ternary mixtures and the quaternary mixture were below the thresholds calculated using Feller’s additive model when the detection probability was 50%, suggesting synergistic effects of these mixtures. Notably, the binary mixing of 2-methylpropanal and benzaldehyde showed a masking effect, but synergistic effects occurred when these were mixed with 2-methylbutanal or 3-methylbutanal. Thus, this revealed the complex perceptual effects of aroma compounds on sensory responses [36,37].

### 2.5. Analysis of Perceptual Interactions of Nutty Aroma Compounds Using σ-τ Diagrams

Although the perceptual interactions of four of the tested compounds have been studied previously, the degrees of synergistic effects when these compounds are present in different concentrations remain unknown. Therefore, we applied the σ-τ plot approach to measure changes in the perceptual intensities of the mixtures according to the concentrations of the components that were altered. In the σ = f (τ) representation, the graph is divided into several regions [19] according to the interaction level (Figure 4a). Most data points for the binary mixtures were in the hypo-addition region. Thus, when the concentration ratio was in this range, the intensity of the mixture was less than the total intensity of the individual components but greater than the intensity of the individual compounds. For all 96 binary mixtures, only the following three were in the optimal additivity region (0.95 < σ < 1.05): (c) 2-methylbutanal and benzaldehyde (τ = 0.22, σ = 0.97, A1B3), (d) 2-methylpropanal and 3-methylbutanal (τ = 0.57, σ = 0.96, A1B3), and (f) 2-methylpropanal and benzaldehyde (τ = 0.39, σ = 1.03, A1B1). These data indicate that these three binary mixtures have no perceptual effect at this concentration. Six mixtures were in the super additive region (σ > 1.05). For example, for the mixture of 2-methylbutanal and 3-mehytlbutanal, one point was located in the hyper-addition region (A1B1 (τ = 0.45, σ = 1.10); Figure 4a). The corresponding concentrations were 175 μg/kg for 2-methylbutanal and 150 μg/kg for 3-methylbutanal. For the mixture of 2-methylbutanal and 2-methylpropanal, two points were located in the hyper-addition region (A1B1 (τ = 0.40, σ = 1.12); A2B1 (τ = 0.55, σ = 1.08); Figure 4b).The corresponding concentrations were 175 and 350 μg/kg for 2-methylbutanal and 150 μg/kg for 2-methylpropanal. These results clearly showed that synergistic effects easily occurred at a low intensity level (generally τ < 0.55) when the concentrations were close to the thresholds. These findings were consistent with those of Wu et al. [20], who reported that enhancements occur when the odor fineness introduced by the threshold odorant helps to bring the overall odor of the mixture closer to the clearly defined odor objective.

## 3. Material and Methods

### 3.1. Chemicals

3-Methylbutanal, 2-methylbutanal, 2-methylpropanal, and benzaldehyde (all chromatographic grade, ≥97% purity) used in the sensory tests, were purchased from Sigma-Aldrich (St. Louis, MO, USA).

### 3.2. Matrix Preparation

The cheese matrix was prepared from fresh cheddar cheese (0 months) purchased from Mengniu Dairy Co., Ltd. (Hohhot, China). The cheese was regarded as having no nutty flavor, as determined by a sensory evaluation. The moisture, fat, protein, salt, and ash contents and pH value of the cheese matrix were determined according to standard methods (Table 3) [38]. The cheese samples were melted by heating at 50 °C in a water bath. Aldehydes were added to the dissolved cheese to prepare a specific concentration gradient, and the resulting mixture was kneaded for 3 min. The cheese matrix was sealed in a bag and tempered overnight in a refrigerator at 4 °C. The cheese matrix was then crushed with a grinder, placed in a tasting cup, and kept at room temperature for 1 h prior to sensory evaluation [39].

### 3.3. Sensory Evaluation

Sensory tests were performed as described in previous studies with some modifications [40,41]. Samples were evaluated at a controlled temperature (20 °C) in individual booths using covered, brown, odorless glasses bottles that contained 5 g of sample and were coded with random three-digit numbers. The evaluation process took 5 min to complete.

Sensory evaluations were performed by 12 panelists (six males and six females with an average age of 23). All panelists belonged to the School of Perfume and Aroma Technology, Shanghai Institute of Technology (Shanghai, China), and had previously received professional training. However, the panelists were not informed about the aim of the experiment. They were selected from among 40 candidates based on their experience in evaluating cheese and their performance in three-alternative forced-choice (3-AFC) tests. They attended five sessions per week for 8 weeks. The same sensory panel participated in all experiments involving sensory evaluations in this study. The specific sensory evaluation experiment was designed with reference to the experimental protocols described in the following sections.

### 3.4. Measurement of Odor Thresholds in the Cheddar Cheese Matrix

The 3-methylbutanal, 2-methylbutanal, 2-methylpropanal, and benzaldehyde thresholds in the cheese matrix were measured using the American Society for Testing and Materials (ASTM) protocol E1432 [42]. Panelists were informed of the nature of the additive, and a standard solution was presented at the entrance to the test room. The threshold values were determined according to the method of Avsar et al. [8]. The panelists performed a number of tests (3-AFC, NF ISO 13301) [43]. The initial concentrations of 2-methylbutanal, 3-methylbutanal, 2-methylpropanal, and benzaldehyde in the cheese matrix were 25, 25, 25, and 50 μg/kg, respectively, based on the results of preliminary experiments. Each session comprised 10 forced-choice tests with increasing concentrations differentiated by a factor of 2.0. Therefore, the concentration ranges of 2-methylbutanal, 3-methylbutanal, 2-methylpropanal and benzaldehyde are 25–12,800 μg/kg, 25–12,800 μg/kg, 25–12,800 μg/kg and 50–25,600 μg/kg, respectively. Each test contained one positive sample supplemented with increasing concentrations of the compound to be evaluated. The olfactory thresholds of 2-methylbutanal, 3-methylbutanal, 2-methylpropanal, and benzaldehyde were measured. All experiments were performed in triplicate.

The results of all 3-AFC tests were statistically analyzed. The olfactory threshold was defined as the concentration at which the probability of detection was 50%. The concentration/response function is a psychometric function that fits a sigmoid curve (y = 1/(1 + e{ − λx})). Detection probability was corrected using the chance factor (P = (3p − 1)/2, where p = the proportion of correct responses for each concentration and P = the proportion corrected for the chance effect, 1/3 for 3-AFC) [23].

### 3.5. Determination of the Optimal Concentration Ranges of Aldehydes in a Cheese Matrix

The optimal concentration ranges of 3-methylbutanal, 2-methylbutanal, 2-methylpropanal, and benzaldehyde were determined using methods described in previous studies, with some modifications [8,44]. The testing concentrations (2-methylbutanal (175–425 μg/kg), 3-methylbutanal (150–400 μg/kg), 2-methylpropanal (150–400 μg/kg), and benzaldehyde (500–3000 μg/kg)) used in this study were based on the results of the threshold study described above. Cheese matrices with different concentrations of aldehydes were prepared according to the method described in Section 2.2, and sensory evaluations were performed according to the method described in Section 2.3. Lightly toasted, unsalted nuts were used as the reference for the nutty aroma, which was scored on a 9-point intensity scale (range: 1 = very low intensity, 5 = medium intensity, 9 = strong intensity). The overall flavor preference was also scored on a 1–9 scale (1 = dislike extremely; 5 = neither like nor dislike; 9 = like extremely) [45]. A series of single-compound samples at various concentrations was presented in a random order. The sensory evaluation team scored the samples at least three times per concentration. The data were summarized as the geometric mean of the scores of all panel members, with standard deviation. Both values > 5 at a specific point indicated as an acceptable concentration by sensory evaluation.

### 3.6. Perceptual Interaction Analysis

#### 3.6.1. Interactions between Aroma Compounds Using the Threshold Approach

Although the threshold of a compound gives a good indication of its flavor impact, it would be insufficient and oversimplified to consider the overall flavor of a cheese as the sum of the contributions made by each individual compound, as several interactions can affect the flavor perception [46]. Therefore, the effects of mixtures of compounds were evaluated by determining the mixture thresholds. Compounds were added in the ratios of their individual thresholds to evaluate the interactions at the same level of sensory activity. The thresholds of the mixtures (THmixt) are expressed as the percentages of their calculated threshold while assuming the independency of the compounds. This concept can also be regarded as the individual threshold of each compound in the presence of the other compound(s), expressed as the percentage of its individual threshold (THind). Four possible interactions may occur. The compounds may exhibit their flavors independently, which means that each compound must be present at 100% of its individual threshold to yield a flavor difference (THmixt = 100% THind). The compounds may also counteract each other (antagonism, THmixt > 100% THind) or exhibit an additive (THmixt = 50% THind) or synergistic effect (THmixt < 50% THind) [32,33].

#### 3.6.2. Perceptual Interactions of Binary, Ternary, and Quaternary Mixtures Determined Using Feller’s Additive Model

The interaction effects for mixtures were evaluated using Feller’s additive model, as adapted by Miyazawa et al. [47]. The determined threshold (Table 1) of the compound, which was premeasured by the same 12 panelists from the sensory evaluation panel, was used as the intermediate concentration. This concentration was then decreased or increased by a factor of two, and 10 suitable sample concentrations were selected to determine the linear range of the fitted curve. Thus, the concentration ranges of 3-methylbutanal and 2-methylbutanal, 2-methylbutanal and 2-methylpropanal, 2-methylpropanal and 3-methylbutanal, 3-methylbutanal and benzaldehyde, 2-methylbutanal and benzaldehyde, and 2-methylpropanal and benzaldehyde are 10.9–5600 μg/kg, 10.9–5600 μg/kg, 9.4–4800 μg/kg, 31.3–16,000 μg/kg, 31.3–16,000 μg/kg, and 31.3–16,000 μg/kg, respectively. The detected proportion of each compound was used to calculate the threshold of each binary, ternary, and quaternary mixture. The probability of detection of the mixture was defined as follows [47,48]:(1)$$P\left(\mathrm{mix}\right)\;=\sum_{i=1}^{n}p\left(i\right)-\sum_{i=1,j=1}^{n}p\left(i\right)p\left(j\right)+\sum_{i=1,j=1,k=1}^{n}p\left(i\right)p\left(j\right)p\left(k\right)-\;\ldots +{\left(-1\right)}^{n-1}\prod_{n}^{i=1}p\left(i\right)$$

In the above formula, I ≠ j … ≠ n. p(i), p(j) … p(n) represent the probability of detecting components i, j …, and n. The measurement probability p(mix) was also determined by the 3-AFC method. The curves for the actual and theoretical models were generated by the data from measured and theoretical p(mix), respectively. Then, the olfactory threshold was defined as the point at which the detection probability was 50%, and the actual and theoretical thresholds for each binary mixture were obtained. If the theoretical threshold for the mixture exceeded the actual, some degree of synergistic or enhanced effect was considered to have occurred. In contrast, if theoretical threshold was below the actual, a certain degree of inhibition was considered to have occurred. Moreover, if the 2 thresholds were equal, we considered there to be no perceptual effects on sensory responses.

#### 3.6.3. σ-τ Plot Analysis

Four concentrations of the four compounds were used to prepare binary mixtures. The selected concentrations were around the respective thresholds (150, 300, 600, and 1200 μg/kg for 3-methylbutanal; 150, 300, 600, and 1200 μg/kg for 2-methylpropanal; 175, 350, 700, and 1400 μg/kg for 2-methylbutanal; and 500, 1000, 2000, and 4000 μg/kg for acetoin). For each pair of compounds, the odor intensity was detected for 16 binary mixtures (4 × 4) and eight single mixtures at different concentrations. Each sample was presented twice during each session. For each sample, the subject rated the intensity of the cheese flavor on a 9 cm scale printed on paper, with 1 and 9 cm corresponding to “no odor perceived” and “very intense,” respectively.

Experimental data on the intensities of binary mixtures are presented graphically (σ = f (τ)), according to the method of Patte and Laffort [49], where τ represents the ratio of the perceived intensity of unmixed A (or B), τA = IA/(IA + IB), and τB = IB/(IA + IB); σ indicates the ratio of the perceived intensity of the mixture to the overall perceived intensities of the individual components of the mixture and reflects the level of interaction: σ = Imix/(IA + IB), where Imix is the perceived intensity of the mixture. Both τ and σ were obtained from the intensity of the overall aroma. The synthetic representation σ = f (τ) reflects the test results. The graph was divided into several parts, according to the interaction level. The location of the experimental data on the graph indicates the interaction level. If the intensity after mixing was the same as the sum of the intensities of individual components before mixing, complete addition was assumed (σ = 1). If the intensity of the mixture was greater than the sum of the intensity of its components, synergy was assumed (σ > 1), but if it was lower than the sum of its components, hypo-addition was assumed (σ < 1). Frijters [50] divided hypo-addition into three stages: “partial addition,” “compromise,” and “subtraction.” If the perceived intensity of the mixture was greater than that of the individual compounds, partial addition was assumed. If the perceived intensity of the mixture was within the range of the individual compound intensities, the mixture was identified as compromise, and if the quality intensity of the mixture was less than that of the individual compounds, it was identified as subtraction. The mean intensity recorded by the 12 subjects was within the 95% confidence interval for statistical tests of σ and τ.

### 3.7. Statistical Analysis

Sigma Plot 12.0 software (SYSTAT, Inc, Chicago, IL, USA) was used to fit Feller’s model, and Origin 9.0 (OriginLab Corporation, Northampton, MA, USA) was used to prepare figures. All statistical analyses were performed using XLSTAT 7.5 (Addinsoft, Long Island City, NY, USA), and Duncan’s test was used to determine statistical differences. A *p*-value < 0.05 was considered statistically significant.

## 4. Conclusions

The present study determined the odor thresholds and optimal concentration ranges of 2-methylbutanal, 3-methylbutanal, 2-methylpropanal, and benzaldehyde in a cheese matrix and further demonstrated the perceptual interactions of these compounds using three different methods. Both the threshold method and Feller’s model showed that five binary mixtures had synergistic effects, whereas the mixture of 2-methylpropanal and benzaldehyde had a masking effect. In addition, the ternary mixtures and quaternary mixture exhibited synergistic effects when assessed using Feller’s additive model. From the σ/τ plot analysis, hypo-addition actions were frequent in the binary mixtures and hyper-addition actions occurred at the threshold concentrations. These findings indicate that perceptual interactions among these nutty aroma compounds in a cheese model vary with different concentrations and threshold ratios. Our findings may lead to a better understanding of the aldehydes responsible for the nutty aromas of cheddar cheese, which would be expected to help achieve desirable organoleptic properties in the final products.

## Figures and Tables

**Figure 1 molecules-25-04308-f001:**
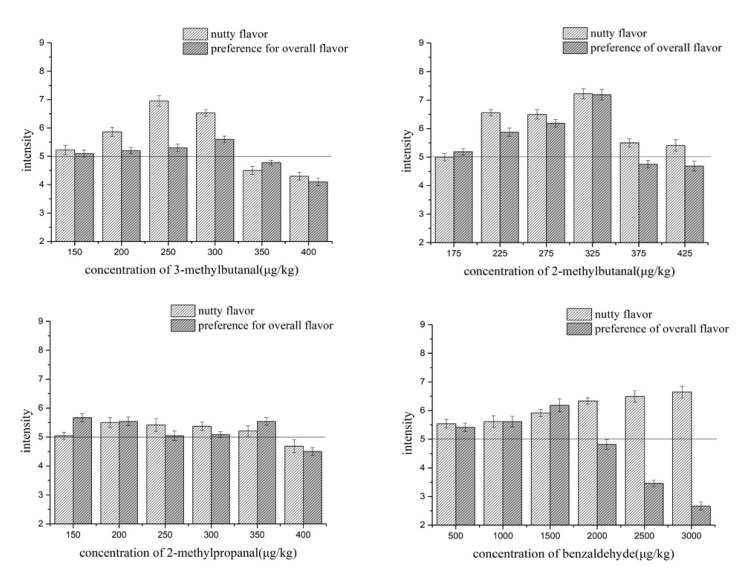
Nutty and overall aroma scores of 3-methylbutanal, 2-methylbutanal, 2-methylpropanal, and benzaldehyde at different concentrations in the cheddar cheese matrix. The horizontal line in the figure indicated the nutty aroma and the overall aroma score was 5, which was regarded as the limit score for acceptable concentration by sensory evaluation.

**Figure 2 molecules-25-04308-f002:**
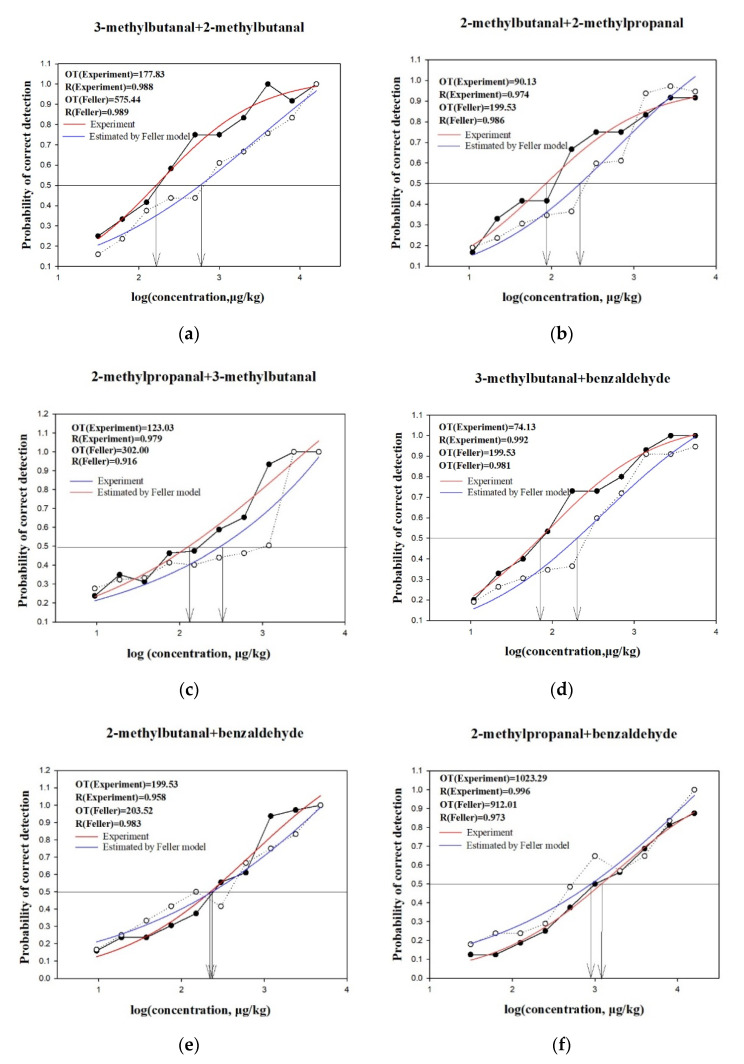
Perceptual interactions among six pairs of binary mixtures as determined using Feller’s additive model. (**a**) 3-Methylbutanal and 2-methylbutanal, (**b**) 2-methylbutanal and 2-methylpropanal, (**c**) 2-methylpropanal and 3-methylbutanal, (**d**) 3-methylbutanal and benzaldehyde, (**e**) 2-methylbutanal and benzaldehyde, (**f**) 2-methylpropanal and benzaldehyde. The horizontal line indicates that the correct detection probability is 50%. The two arrows represent the corresponding log (concentration) values under the correct detection probability of 50%. The solid dots represent the detection probability obtained by the experiments, and the empty dots represent the detection probability obtained by the Feller model. OT, olfactory threshold. R—Feller’s additive model fitting result.

**Figure 3 molecules-25-04308-f003:**
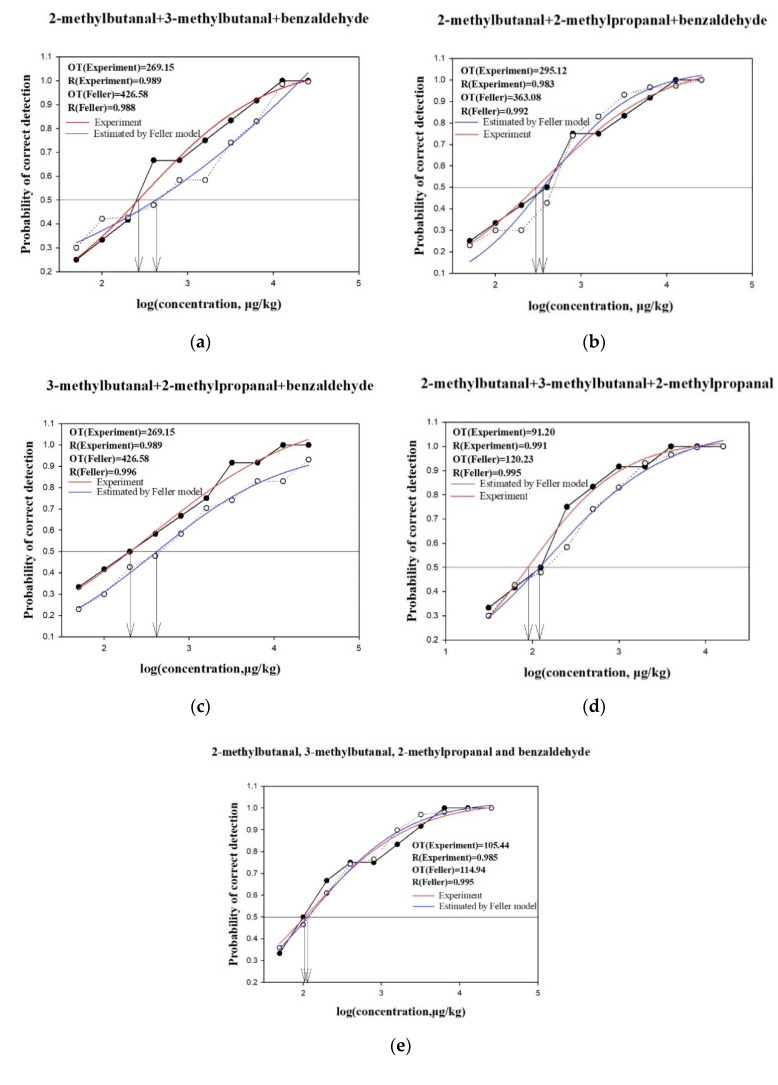
Perceptual interactions among four ternary mixtures and one quaternary mixture as determined using Feller’s additive model. (**a**) 2-Methylbutanal, 3-methylbutanal, and benzaldehyde; (**b**) 2-methylbutanal, 2-methylpropanal, and benzaldehyde; (**c**) 3-methylbutanal, 2-methylpropanal, and benzaldehyde; (**d**) 3-methylbutanal, 2-methylbutanal, and 2-methylpropanal; (**e**) 2-methylbutanal, 3-methylbutanal, 2-methylpropanal, and benzaldehyde. The horizontal line indicates that the correct detection probability is 50%. The two arrows represent the corresponding log (concentration) values under the correct detection probability of 50%. OT, olfactory threshold. R—Feller’s additive model fitting result.

**Figure 4 molecules-25-04308-f004:**
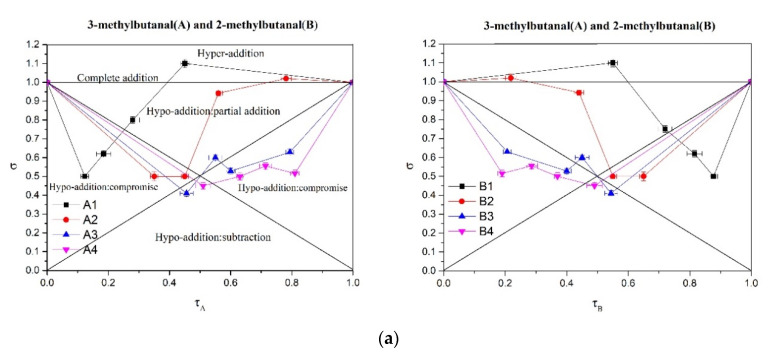
σ-τ diagram representations of seven pairs of binary mixtures. τ refers to the ratio of the perceived intensity of the binary mixture to the sum of the perceived intensities of the individual components before mixing. σ refers to the ratio of the perceived intensity of the mixture to the sum of the perceived intensities of each single component and reflects the level of various interactions. (**a**) 3-methylbutanal and 2-methylbutanal, (**b**) 2-methylbutanal and 2-methylpropanal, (**c**) 2-methylbutanal and benzaldehyde, (**d**) 2-methylpropanal and 3-methylbutanal, (**e**) 3-methylbutanal and benzaldehyde, (**f**) 2-methylpropanal and benzaldehyde pair. A1–A4 represents four concentrations of compound A from low to high. B1–B4 represent four concentrations of compound B from low to high. Error bars indicate 95% confidence intervals of the means for both τ and σ.

**Table 1 molecules-25-04308-t001:** Detection thresholds of 3-methylbutanal, 2-methylbutanal, 2-methylpropanal, and benzaldehyde.

Compound	Aroma Description	Threshold (Determined, μg/kg)	Threshold in Water (Literature, μg/kg)	Ratio (Determined/Literature)
3-Methylbutanal	Malty, nutty, almond, cocoa	150.3	1.1	136.7
2-Methylbutanal	Malty, almond, cacao, apple-like	175.4	1.0	175.4
2-Methylpropanal	Pungent, varnish, fruity	150.7	1.5	100.2
Benzaldehyde	Almond, cherry stone	500.2	350.0	1.4

**Table 2 molecules-25-04308-t002:** Threshold values of mixtures of compounds in the ratios of their individual thresholds (TH) and the corresponding flavor descriptions.

9	2 Compounds	Flavor Description	TH of Mixture ^a^ (%)	SD ^b^
1	2-methylbutanal + 3-methylbutanal	Nutty, malty	29.29%	1.90
2	2-methylbutanal + 2-methylpropanal	Milky, malty	49.08%	2.10
3	2-methylbutanal + benzaldehyde	Milky, almond	46.59%	2.50
4	3-methylbutanal + 2-methylpropanal	Flowery, caramel, fruity, malty	40.90%	3.50
5	3-methylbutanal + benzaldehyde	Nutty, milky	38.36%	5.60
6	2-methylpropanal + benzaldehyde	Malty	148.20%	10.30

^a^ Threshold values of mixtures of compounds in the ratio of their individual thresholds. ^b^ The standard deviation of thresholds of the mixtures’ compounds.

**Table 3 molecules-25-04308-t003:** Basic information and chemical composition of the cheddar cheese matrix.

Sample	Origin	Maturity	Moisture%	Fat%	Protein%	Salt%	Ash%	pH
**Fresh cheese**	China	0 month	43.21 ± 0.67	22.10 ± 0.91	28.01 ± 0.63	1.34 ± 0.25	3.38 ± 0.19	5.29 ± 0.00

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
