# Peer review of "Evaluation of the Perceptual Interactions among Aldehydes in a Cheddar Cheese Matrix According to Odor Threshold and Aroma Intensity"

_molecules, 2020, doi:10.3390/molecules25184308_

Round 1
Reviewer 1 Report
This paper describes contributions of selected short-chain aldehydes in Cheddar cheese. Finally, an interesting synergistic olfactory effect has been found.
1. Please precise the solvent (if was used) for aldehydes added in experiment (L. 91-92); 2. Does ash in cheese include salt? 3. For Table 2, please decrease the significant digits for Threshold Determined as well as Ratio Determined/Literature; 4. For Threshold Literature please add solvent used; 5. For better visualisation of results on Fig. 1 I advise re-scale charts for linear aldehydes (e.g. from value 3 or 4); 6. For Fog. 2, please add info (legend) about solid and empty dots (model and experiment); 7. Experimental concentration range presented on Fig. 2 (up to around 5000 milligrams/kg do not correspond with those presented and described in materials and methods as well as previous parts.
Author Response
Dear Reviewer,
On behalf of the co-authors, I would like to thank you and the reviewers for the detailed review and comments on this manuscript. We have revised the manuscript accordingly and highlighted the revised words and phrases, and the full list of responses to the reviewers’ comments is enclosed.
Thank you again for your comments. Look forward to your decision.
Sincerely yours,
Chen Chen
Department of Food Science and Technology, Shanghai Institute of Technology, 100 Haiquan Road,
Shanghai 201418, China.
E-mail: chenchen@sit.edu.cn

Reviewer 2 Report
The present paper determined odor thresholds and optimal concentration ranges of 2-methylbutanal, 3-methylbutanal, 2-methylpropanal, and benzaldehyde, known to contribute to "nutty" flavor, in a cheese matrix. It further demonstrated the perceptual interactions of these compounds using three different methods. Both synergistic and masking effects were observed, mainly with regard to different concentrations and threshold ratios.
The paper is well written, some minor corrections are indicated below. Introduction includes all appropriate information and relevant scientific background, at an excellent extent. Material and Methods section needs small clarifications that are listed below. Results and discussion section, while with adequate results' presentation, needs to be enriched with more "discussion" in order to help the reader better understand the relation between the aim of the study, the results obtained and the existing literature.
line 47-48: thresholds of 2-methylbutanal, 3-methylbutanal, 2-methylpropanal, and benzaldehyde are also determined in model wine solution (ex. Culleré, L., Cacho, J., & Ferreira, V. (2007). An assessment of the role played by some oxidation-related aldehydes in wine aroma. Journal of Agricultural and Food Chemistry, 55(3), 876-881)
line 84: correct "use" to "used"
line 117: "increasing concentrations" : put concentration range
line 164: 'estimated threshold": in the table we see "determined threshold"
line 167-168: what is "detected proportion" of each compound?
line 246: correct "consumers" to "panellists"
general remark for 242-243: I don't agree that "The results of 2-methylbutanal and 2-methylpropanal were similar to those of 3-methylbutanal." In my opinion, 2-methylbutanal was higer in preference with values going up to 7.0, while the other two volatiles hardly exceeded value of 5 = neither like nor dislike, which is rather neutral.
line 256: correct "normal conditions" to "model cheese conditions"
line 467: correct the names in the reference to correspond with the paper
line 471: correct the names in the reference to correspond with the paper
Author Response

(The authors gave the same response as above.)
